# Latest Advances in Arbovirus Diagnostics

**DOI:** 10.3390/microorganisms11051159

**Published:** 2023-04-28

**Authors:** Jano Varghese, Imesh De Silva, Douglas S. Millar

**Affiliations:** Genetic Signatures, 7 Eliza Street, Newtown, Sydney 2042, Australia; jano.varghese@geneticsignatures.com (J.V.); imesh.desilva@geneticsignatures.com (I.D.S.)

**Keywords:** isothermal amplification, arboviral diagnostics, point of care, decentralized testing

## Abstract

Arboviruses are a diverse family of vector-borne pathogens that include members of the *Flaviviridae*, *Togaviridae*, *Phenuviridae*, *Peribunyaviridae*, *Reoviridae*, *Asfarviridae*, *Rhabdoviridae*, *Orthomyxoviridae* and *Poxviridae* families. It is thought that new world arboviruses such as yellow fever virus emerged in the 16th century due to the slave trade from Africa to America. Severe disease-causing viruses in humans include Japanese encephalitis virus (JEV), yellow fever virus (YFV), dengue virus (DENV), West Nile virus (WNV), Zika virus (ZIKV), Crimean–Congo hemorrhagic fever virus (CCHFV), severe fever with thrombocytopenia syndrome virus (SFTSV) and Rift Valley fever virus (RVFV). Numerous methods have been developed to detect the presence of these pathogens in clinical samples, including enzyme-linked immunosorbent assays (ELISAs), lateral flow assays (LFAs) and reverse transcriptase–polymerase chain reaction (RT-PCR). Most of these assays are performed in centralized laboratories due to the need for specialized equipment, such as PCR thermal cyclers and dedicated infrastructure. More recently, molecular methods have been developed which can be performed at a constant temperature, termed isothermal amplification, negating the need for expensive thermal cycling equipment. In most cases, isothermal amplification can now be carried out in as little as 5–20 min. These methods can potentially be used as inexpensive point of care (POC) tests and in-field deployable applications, thus decentralizing the molecular diagnosis of arboviral disease. This review focuses on the latest developments in isothermal amplification technology and detection techniques that have been applied to arboviral diagnostics and highlights future applications of these new technologies.

## 1. Introduction

Arboviral disease in humans can range from asymptomatic to life-threatening conditions, such as hemorrhagic fevers and encephalitis. Arboviruses are distributed worldwide [1] with some viruses showing restricted geographical distribution (Figure 1). However, as a result of environmental destruction, the travel boom, deforestation, urbanization and failure of vector control programs, arboviruses have expanded into areas not previously seen [2]. Due to the severity and global distribution of arboviral disease, it is vital to have sensitive and rapid diagnostics tests available to determine the causative agent responsible for infection and the implementation of control strategies. Traditionally, serological methods have been the mainstay for the diagnosis of arboviral disease. Methods such as direct and indirect enzyme-linked immunosorbent assays (ELISAs) and lateral flow assays (LFAs) have been widely used to diagnose many arboviral diseases, including Zika, dengue, chikungunya and yellow fever virus. More recently, next-generation sequencing (NGS) approaches have been applied for the detection of arboviruses, including Western bluetongue virus [3], chikungunya, Zika [4], West Nile virus [5] and, in some cases, NGS methods, which have been shown to have a sensitivity similar to that of conventional RT-PCR. Although NGS is well suited for surveillance approaches, these methods are more time consuming and costly, and they require dedicated equipment and computing networks. They are also unsuitable for routine screening when compared to traditional methods. RT-PCR assays tend to be carried out at centralized testing facilities due to the need for thermal cycling systems and related equipment. Advances in molecular biology have led to the discovery of several isothermal nucleic acid amplification technologies (INAATs) that can amplify nucleic acids at a constant temperature without the need for expensive thermal cycling equipment. These newer approaches offer the prospect of decentralizing molecular diagnostic testing and the potential of rapid, cheap point of care (POC) screening and in-field testing applicable to resource-limited settings.

Over the last 150 years, a number of notable epidemics and pandemics have occurred. The influenza A H1N1 pandemic of 1918–1920 resulted in an estimated 50–100 million deaths, the HIV pandemic beginning in the 1980s has affected over 40 million people, and the SARS-CoV-1 epidemic of 1983 and the swine flu pandemic starting in 2009 have caused significant morbidity and mortality. The Ebola outbreaks of 2014–2016 resulted in over 11,000 fatalities, and the Zika outbreaks in 2015–2016 caused a substantial burden of disease globally [6]. Finally, the SARS-CoV-2 pandemic beginning in 2019 has infected nearly 700 million individuals, resulting in nearly 7 million deaths. These outbreaks highlight the importance of novel diagnostics and POC devices that can detect infectious agents in an accurate and timely manner as they arise.

Arboviruses will continue to emerge and re-emerge over time, a notable example being Zika virus which, until recent epidemics, was considered a virus that caused relatively mild infection in humans, but it has since been shown to cause microcephaly and Guillain–Barré syndrome [7]. It has been suggested that one of the more obscure viruses of the *Flaviviridae* family, such as Spondweni virus (SPOV), Usutu virus (USUV), Ilheus virus (ILHV), Rocio virus (ROCV) and Wesselsbron virus (WSLV), or one of the tick-borne family of flaviviruses could emerge into the human population and cause significant health concerns [7]. Rift Valley fever virus (RVFV) may be one of the next *Phleboviruses* to emerge as an important human threat due to its continued geographical spread [8]. Alphaviruses such as Mayaro virus (MAYV), which is native to the Americas, may over time adapt to different mosquito populations, such as *Aedes*, and emerge as a more significant human pathogen [9]. Many arboviruses are found in resource-limited settings that in some cases have inadequate infrastructure for diagnostic testing, emphasizing the importance of inexpensive, rapid and sensitive POC tests that can be used for field deployment.

**Figure 1 microorganisms-11-01159-f001:**
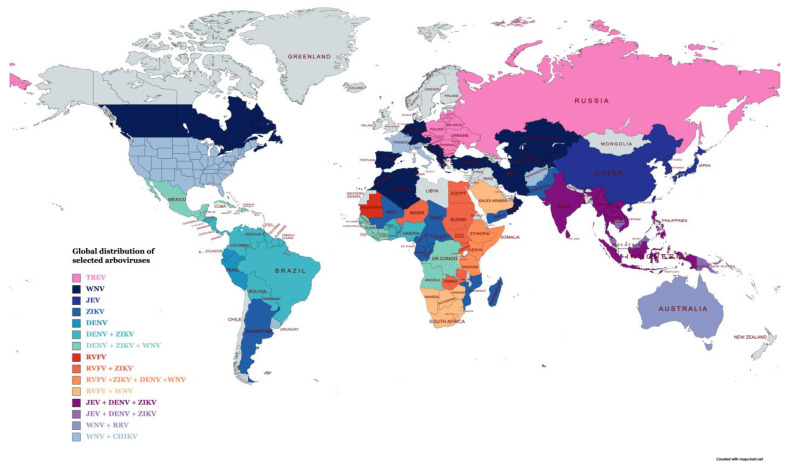
The global distribution of a number of important arboviruses (this map was prepared using information in Socha et al. [1] and references [10,11,12,13,14,15,16,17,18,19,20,21,22,23,24,25,26,27,28,29,30,31,32,33] using the free web-based MapChart software). Table legend abbreviations: tick-borne encephalitis virus (TBEV; West Nile virus (WNV); Japanese encephalitis virus (JEV); Zika Virus (ZIKV); dengue virus (DENV); Rift Valley fever virus (RVFV); Ross River virus (RRV); and chikungunya (CHIKV).

## 2. Arbovirus Disease in Humans and Animals

Arboviruses consist of a diverse family of pathogens that can infect a wide range of animals and humans. Arboviruses are predominantly positive or negative single-stranded or double-stranded RNA containing viruses from the *Flaviviridae*, *Togaviridae*, *Phenuviridae*, *Peribunyaviridae*, *Reoviridae*, *Rhabdoviridae*, *Orthomyxoviridae* and *Gammaentomopoxvirus* families. The only significant DNA-containing virus is the African swine fever virus that belongs to the *Asfarviridae* family. The 1992 International Catalogue of Arboviruses registered 535 species of virus belonging to 14 families; however, this number is continually increasing due to improvements in isolation and molecular methods for virus discovery and surveillance [34].

Arboviruses must infect their insect vector prior to transmission to a susceptible host [35]. Arboviruses are generally spread as a result of a bite from infected mosquitoes, ticks or other biting flies. Arboviruses circulate among wild animals and birds and are then transmitted as a spill over to humans and domestic animals, which are dead end hosts [36]. Humans and animals infected with arboviruses can display a wide range of symptoms from asymptomatic to life-threatening conditions, such as hemorrhagic fevers and encephalitis, which can often result in long-term complications [8].

A significant number of arboviruses cause human disease and the morbidity and mortality associated with infection cause a substantial social and economic burden when outbreaks occur. Table 1 illustrates a number of arboviruses capable of causing disease in humans. These viruses are distributed on a global scale with some viruses restricted to specific geographical locations corresponding to the distribution of their insect vectors. Mosquitoes are responsible for the transmission of many emerging and re-emerging arboviruses, including the four serogroups of dengue, chikungunya, yellow fever and Zika viruses [37]. These viruses cause a severe burden of disease, with up to 400 million infections and 100 million clinical cases of dengue recorded in 2010 [38].

## 3. Arbovirus Diagnostic Approaches

### 3.1. Serology

Traditionally, diagnosis of arbovirus infection has been performed by serology. Serological methods include direct test involving virus detection and indirect tests targeting antibodies produced during viral infection. One of the most common techniques used for arboviral diagnosis is enzyme-linked immunosorbent assays, such as (ELISA)-IgM capture, which targets antibodies produced early in response to infection and indirect IgG assays. However, due to cross-reactivity with closely related viruses, confirmatory assays, such as the plaque reduction neutralization test (PRNT), which quantifies neutralizing antibodies in serum or CSF samples, are required. Other techniques, such as immune magnetic agglutination assays and rapid diagnostic tests (RDTs), including lateral flow assays, have been used extensively in arbovirus diagnostics [39]. Serology has been applied to the detection of many arboviruses, and commercial kits are available for dengue [MyBiosource, San Diego, CA, USA], Zika [Vircell, Valencia, Spain], chikungunya [Abcam, Cambridge, UK], yellow fever virus [Abbexa, Cambridge, UK], Rift Valley fever virus [Immune technology, New York, NY, USA] and many more.

Although serological methods are generally quick and inexpensive, this technology has several drawbacks. Cross-reactivity between related viruses can cause false positive results and this has been noted with Zika, dengue, West Nile virus and Japanese encephalitis virus [40]. Secondly, immunocompromised patients with impaired B-cell responses may have false negative serology results [41]. Another limitation of indirect antibody methods is the window period of between 3–5 days after initial infection before antibodies can be detected, potentially resulting in false negative results during early infection. Using direct methods, such as NS1 antigen assays for dengue, can overcome this drawback; however, the sensitivity of these methods can be low [42]. Furthermore, no serotype information can be obtained using this method, necessitating molecular approaches to distinguish the individual virus responsible for disease, which is essential for patients suffering from repeat dengue infection.

### 3.2. Next-Generation Sequencing (NGS)

NGS allows for unbiased sequencing of clinical samples and vector pools, which can be helpful in surveillance programs where an unknown pathogen is suspected. Viral metatranscriptomic next-generation sequencing (mNGS) techniques have been used to screen vectors for arboviruses in Australia during adverse weather events, favoring vector activity detecting Ross River virus (RRV), Sindbis virus (SINV), Trubanaman virus (TRUV), Umatilla virus (UMAV), and Wongorr virus (WGRV). The use of NGS in this situation enabled the discovery of multiple virus types in the pools, which would have been more challenging using a targeted RT-PCR approach [43].

This method has been used to identify arboviruses in patients with undiagnosed encephalitis/meningitis and in patients with symptoms consistent with arboviral infection but yielding inconclusive RT-PCR data [5,44,45,46]. NGS has proved to be a useful tool to analyze strains of *Orthomyxoviridae* obtained from different geographical areas, resulting in the classification and separation of these viruses into two major clades, namely the Thogoto-like and Dhori-like viruses [47].

In a survey of acute febrile illness in Columbia, samples were initially screened by RT-PCR and rapid tests for DENV, ZIKV and CHIKV, followed by testing negative samples with mNGS, resulting in the discovery of undetected Oropouche virus (OROV) circulation. Interestingly, the study confirmed OROV as an emerging pathogen in the area using RT-PCR, finding that virus activity was associated with specific locations, climate and clinical symptoms [48]. mNGS has been applied for the detection of Orbivirus in mosquito vectors from Japan [49] and mosquito species’ identification in Mexico [50].

While mNGS is primarily a tool used to monitor the presence of viruses from clinical and environmental samples, which could result in significant public health threats, methods have been developed for the enrichment of specific virus sequences in samples such as vector pools. Viral specific enrichment was successfully used to characterize Crimean-Congo hemorrhagic fever virus (CCHFV) and Jingmen virus (JMTV) in a pooled tick sample [51] and in an arbovirus surveillance program in Australia that detected unknown Ross River virus circulation [52].

Although the costs and labor involved in NGS protocols has dropped dramatically, NGS is still relatively costly, requires extensive equipment and computer networks, as well as highly trained staff, and is unsuitable for use in POC applications and in resource-limited settings.

### 3.3. Reverse Transcriptase PCR (RT-PCR)

RT-PCR has been used for many years in arbovirus diagnostics and has the advantage of increased sensitivity and specificity compared to serological and NGS methods without prior viral enrichment strategies. RT-PCR has become the gold standard for diagnosing many viral diseases, including respiratory, gastrointestinal, sexually transmitted, meningitis and tropical diseases. RT-PCR has been used to diagnose a wide range of both human and animal arboviral infections.

Generally, RT-PCR is not compatible with point of care (POC) applications, mainly due to the requirement for thermal cycling equipment, infrastructure and trained staff [53]. Novel RT-PCR technology based on the bisulphite conversion of cytosine to thymine has been developed, which simplifies conventional nucleic acid sequences containing adenine, thymine, guanine and cytosine to a 3base^TM^ form consisting of adenine, thymine and guanine (Genetic Signatures, Newtown, NSW, Australia). After conversion to a 3base™, genome members of viral families become more similar at the nucleic acid level (Table 2). Thus, pan-family PCR primers and probes that contain fewer mismatched sequences after conversion can be designed, resulting in more efficient amplification. This technology allows laboratories to detect the presence of arboviral families, such as flavivirus and alphavirus, using a pan-family approach during disease outbreaks and would be ideally suited to viral surveillance approaches [54].

#### Sample-to-Answer RT-PCR Systems

To produce RT-PCR assays compatible with POC applications, several sample-to-answer devices have been developed, including the GenXpert™ system (Cepheid, Sunnyvale, CA, USA), FilmArray^®^ (Biofire, Salt Lake City, UT, USA), Sal6830 MicroGEM (Dale Avenue, Charlottesville, VA, USA) and the Cobas™ Liat™ system (Roche Diagnostics, Indianapolis, IN, USA). These systems are all-in-one devices that simultaneously extract and purify nucleic acids from clinical samples, perform PCR amplification and yield a positive/negative read-out at the end of the reaction. Typical run times are 30–60 min, and sensitivity can range from as low as 12 to 6.4 × 10^3^ copies/mL [55]. Veredus laboratories (Science Park Drive, Singapore) have developed a RT-PCR-based POC system for research use only, VereFever™, which can diagnose the presence of CHIKV, DENV 1–4, JEV, WNV, YFV and ZIKV on a single chip, making it the most comprehensive commercially developed system for arbovirus diagnostics. Although these systems are promising for POC and fieldwork applications in general, they are expensive, with reagent costs greater than USD 100 per sample.

## 4. Isothermal Amplification Technologies

### 4.1. Strand Displacement Amplification (SDA)/Nicking Endonuclease Amplification Reaction (NEAR)

Walker et al. developed the original strand displacement amplification reaction in 1992. This method relies on generating a hemi-modified restriction site on one DNA strand and the 5′-3′ exonuclease activity of a DNA polymerase. A specific restriction enzyme nicks the DNA, and the strand is then displaced and replaced with a newly copied strand via the strand displacement/polymerase activity of the DNA polymerase. The reaction continues at a single temperature, without the requirement of thermal cycling, leading to exponential amplification of the DNA or RNA target [56]. The method was subsequently modified [57], and Figure 2a shows the simplification of the technique, resulting in 10^7^-fold amplification efficiency within 60–120 min. Recently, novel technology was developed to amplify RNA in 10–13 min. This method, nicking endonuclease amplification reaction (NEAR), was developed by Abbott laboratories and used in a sample-to-answer device that detected the presence of SARS-CoV-2 in nasopharyngeal swabs. The reaction is carried out in a small lightweight portable instrument requiring minimal hands-on time and training. The technique is capable of a 10^9^ amplification efficiency of starting material in less than 15 min, making it ideal for field-based or POC applications [58].

### 4.2. Nucleic Acid Sequenced-Based Amplification (NASBA)

One of the first technologies developed for the isothermal amplification of nucleic acids, nucleic acid sequence-based amplification (NASBA), was developed in 1991. This method employs a three-enzymes approach using AMV reverse transcriptase, RNase H and T7 RNA polymerase to efficiently amplify RNA target sequences [59]. Initially, a primer complimentary to the RNA of interest containing a T7 polymerase promoter tail binds to the RNA and is copied by AMV reverse transcriptase to produce a complementary DNA (cDNA) strand. The RNA strand is then degraded by the action of RNase H to produce single-stranded DNA. A second primer then binds to the cDNA and is copied to produce a double-stranded DNA molecule. T7 RNA polymerase then binds to the T7 promoter tail and produces a new RNA strand. The process then proceeds exponentially to produce multiple RNA copies of the original target sequence. The method is carried out at 41 °C, without the need for thermal cycling equipment, and is capable of a 10^6^–10^9^ amplification efficiency in 60–90 min. The NASBA reaction produces RNA molecules complementary to the original target RNA sequence. NASBA amplification products can be detected using gel electrophoresis, fluorescence probes with real-time NASBA and colorimetric techniques, such as NASBA-ELISA.

### 4.3. Helicase Dependant Amplification (HDA)

HDA is an isothermal amplification technique that relies on the endogenous properties of a helicase enzyme, and several accessory proteins to unwind double-stranded DNA to generate single-stranded DNA templates for primer binding. HDA is carried out at a constant temperature, removing the requirement of thermal cycling equipment required by conventional PCR methods. After primer binding, the separated strands are copied using a DNA polymerase, resulting in exponential amplification of the target sequence [60] as shown in Figure 2b. Typically, HDA reactions are carried out at between 37 °C to 65 °C, depending on the choice of enzyme, and are capable of a 10^6^-fold amplification efficiency in 30–90 min. HDA has been adapted for colorimetric, fluorescent and lateral flow read-outs, making it applicable to POC and in-field applications [61]. HDA has been used to detect a large number of human and animal pathogens, including Norovirus [62], SARS-CoV-2 [63], *M. tuberculosis* [64], *T. vaginalis* [65] and yellow fever virus in a resource-limited setting [66].

### 4.4. Loop Mediated AMPlification (LAMP)

The LAMP technique first described in 2000 [67] is a method used to amplify either DNA or RNA at a constant temperature using Bacillus stearothermophilus DNA Polymerase I (Bst), which has both DNA polymerase and strand displacement activity. The method relies on the continuous synthesis and displacement of copied DNA without the need to denature the strands with heat, negating the need for thermal cycling equipment. The method uses 4–6 primers specific to the target region of interest. During the initial stages of the reaction, all primers are used, while in the later stages, only the inner primers are responsible for exponential amplification (Figure 2c). The original paper described a 10^9^ amplification efficiency from low copy starting material in as little as 60 min [67]. Typically, LAMP reactions are performed at a temperature of 65 °C, the optimal temperature for Bst DNA polymerase activity. Technology is currently being developed for next-generation polymerase enzymes capable of amplifying nucleic acids at ambient temperatures [68]. The use of ambient temperature amplification of nucleic acids combined with INAAT would greatly simplify both POC and field-based detection strategies.

LAMP is the one of the most common isothermal methods used as the basis for molecular POC devices [69,70]. LAMP has a number of advantages when compared to RT-PCR, the most important of which is the ability to perform the reaction at a single fixed temperature. In addition, a positive reaction can be read by the naked eye using color change or turbidity [71]. Another advantage is the speed at which the reaction occurs, which has been demonstrated to be less than 15 min for the detection of Ebola virus infection [72]. LAMP enzymes and reagents used to perform the reaction are more resistant to inhibitors commonly found in biological samples compared to Taq polymerase [73]. For these reasons, LAMP lends itself to the type of POC and in-field applications that would be ideal for arbovirus diagnosis in remote and resource-limited settings.

A large number of read-out methods have been developed to visualize the LAMP reaction. The first of these was by turbidity which, as a result of the large amounts of amplification product generated during the reaction, results in the accumulation of pyrophosphate, which turns the reaction cloudy and can be observed by the naked eye [74]. Colorimetric detection can be used by adding hydroxy napthol blue, resulting in a color change from violet to sky blue [75]. Another approach uses a change of color from pink to yellow that occurs in positive samples using the NEB WarmStart^®^ colorimetric master mix (https://www.nebiolabs.com.au, accessed 15 April 2023). Fluorescent probes have been developed for the detection of positive samples, opening the possibility of multiplexing the LAMP reaction [76].

One notable method that can be coupled with the LAMP reaction is the CRISPR-Cas12 enzyme complex (see Section 6.5), which can be read using lateral flow [77], microfluidic [78] and fluorescent-based methods [79]. The use of CRISPR-Cas12 technology is ideally suited for rapid field deployable applications.

**Figure 2 microorganisms-11-01159-f002:**
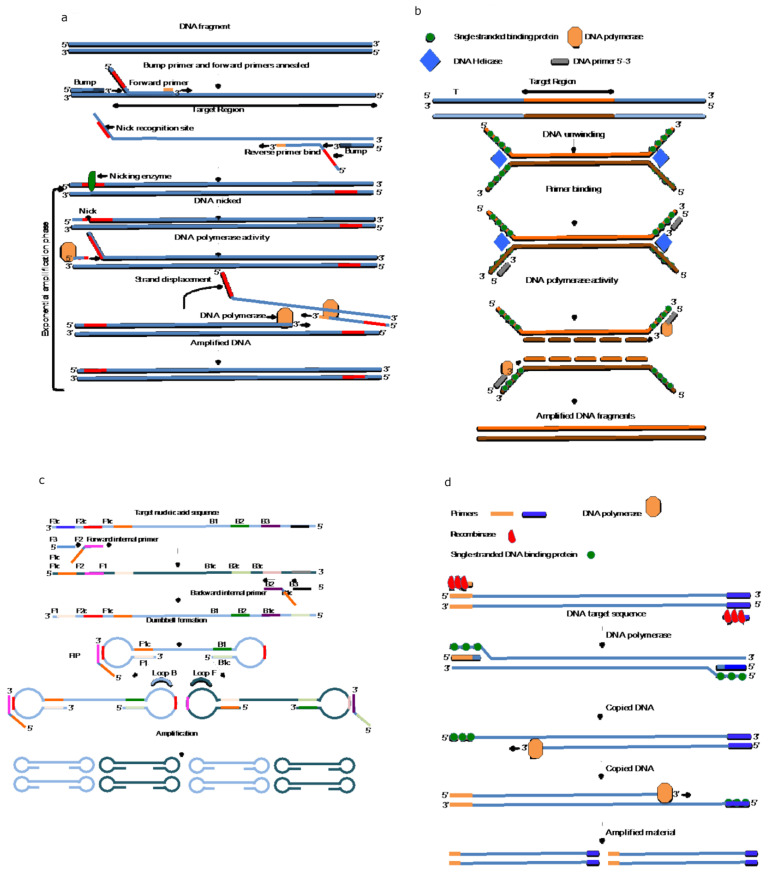
Representation of a number of different isothermal amplification techniques. (**a**) shows the mechanism of action of strand displacement amplification, (**b**) shows helicase-dependent amplification, (**c**) LAMP and (**d**) recombinase polymerase amplification. The figure was generated using standard computer software based on the information contained in references [56,60,67,80].

### 4.5. Recombinase Polymerase Amplification (RPA)/Recombinase Aided Amplification (RAA)

The RPA method was described in 2006 as a method that used a recombinase-primer complex to induce strand exchange at primer recognition sites. This leads to a destabilization of the DNA duplex, and the interaction with single-stranded binding proteins results in strand displacement. DNA is then copied using strand displacing/DNA polymerase enzyme Bst DNA polymerase I, which is also used in the LAMP reaction [80] as shown in Figure 2d. This method is compatible with lateral flow and fluorescent detection of the amplified fragments.

Like other isothermal methodologies, no thermal cycling is required to denature the copied DNA strands. RPA is carried out at temperatures between 37 °C and 42 °C and has been shown to result in a 10^7^–10^8^ amplification efficiency in as little as 60 min. Recent improvements to the technology have resulted in amplification times as low as 3–15 min, making it suitable for rapid POC and mobile field deployable use [81]. Like LAMP, the RPA method is more tolerant of inhibitors in the primary samples; thus, in some instances, it can negate the need for complex sample processing methods in patient samples and arthropod vectors [73]. RPA has been used for the detection of pathogens, including Rift Valley fever virus [82], *F. tularensis* [83], HIV [84], biothreat agents [85] and yellow fever virus [66].

RPA has been coupled with the CRISPR-Cas12 system and a lateral flow or fluorescent read-out for the diagnosis of West Nile virus, Zika and all four dengue subtypes [86].

## 5. Isothermal Amplification and Arbovirus Diagnostics

As previously mentioned, RT-PCR methods have become the gold standard for the detection of many infectious agents. RT-PCR is the most used molecular method for the diagnosis of arboviral disease. LAMP, RPA, SDA and NASBA have all been applied to arbovirus diagnoses and have the advantage of being applicable in resource-limited and POC settings.

Table 3 provides a summary of the major features of isothermal amplification techniques compared to conventional RT-PCR. RT-PCR is a well-established method for the targeted amplification of nucleic acids and has proven to have a high level of specificity and sensitivity. The major drawback of this method is the requirement of a thermal cycler that, in most cases, needs to be fitted with sensitive optical filters to detect the fluorescence generated during the reaction. RT-PCR also requires trained laboratory staff to perform the reaction and, in most cases, dedicated infrastructure. The main advantage of all INAAT methods is that single temperature amplification removes the need for expensive thermal cycling equipment.

Each INAAT method performs amplification in a unique manner that ultimately results in exponential amplification of the nucleic acid sequences of interest. NASBA is the only method that is specific for RNA sequences, while the other methods can be used with both RNA and DNA templates. NASBA, HDA and RPA, similar to RT-PCR, require only two primers for the amplification of target sequences, whereas SDA requires a further two outer (bump) primers to initiate the reaction. The LAMP method is the most complex, requiring 4–6 primers, which can be complicated to design without specific design programs. LAMP primers can be designed using publicly available sites such as https://lamp.neb.com/#!/ (accessed 15 April 2023), which greatly simplifies the process.

INAAT has the advantage that amplification can be carried out very rapidly with common times of 5–30 min using the LAMP, RPA/RAA and NEAR methods. Rapid amplification is important in POC and for field deployable use; thus, these methods are an ideal choice for such applications. In addition, most of these technologies have been adapted so they can be coupled with relatively inexpensive read-outs such as LFDs for use in resource-limited settings. One drawback of some INAAT is that due to the large amounts of product produced during the reaction, non-specific amplification can occur especially with LAMP. The SDA technique, although an isothermal method, requires the DNA strands to be denatured thermally at the beginning of the reaction and has the lowest sensitivity of the methods described.

## 6. Point of Care Diagnostics

The World Health Organization (WHO) has released guidelines that define the desirable properties of POC devices for resource-limited regions. These guidelines led to the ASSURED and (RE)ASSURED criteria [87,88]. To qualify, the method must be at least Affordable with good Sensitivity and Specificity, is User friendly, produces Rapid results with minimal Equipment required and is Deliverable to the end users.

INAAT techniques are ideally suited for adaptation to POC and field deployable settings [89]. A wide range of technologies have been developed as promising candidates for POC devices which could be used in combination with INAAT and are briefly discussed below.

### 6.1. Lab on a Chip

Lab on a chip (LOC) devices perform all steps from sample purification to detection and result read-out. These devices are usually fully integrated, relying on microfluidics with all reagents for sample testing loaded onboard the instrument or in cartridges that can be easily inserted into the platform [90]. LOC devices have been used for the diagnosis of several arboviral diseases. Velders et al. developed a simple and cheap battery-operated device that could be used in the field to detect Zika virus [91]. Sharma et al. developed a microfluidic platform that used magnetic beads coupled with LAMP to detect Zika virus in around 40 min [92]. Song et al. developed a simple disposable microfluidics cassette that could be used to detect Zika in less than 40 min from saliva samples [93]. Finally, Ganguli et al. developed a microfluidics card coupled with LAMP using dried reagents combined with a smartphone read-out to detect Zika, dengue and chikungunya from whole blood samples [94]. LOC devices are therefore ideally suited to diagnosing arboviruses in resource-limited settings, field testing and POC applications.

### 6.2. Lab on a Disc (LOAD)

Like LOC devices, LOAD instruments perform all steps of sample preparation, amplification and result read-out. The difference between the two methods is that LOAD utilizes centrifugal forces, which can be coupled with microfluidics [90] and detection reagents. An example of a commercially available LOAD system is LIASON^®^MDX from Diasorin, which has been used for the detection of Bordetella, *Clostridium difficile*, Influenza, SARS-CoV-2, CMV, HSV, VZV and dengue (Diasorin, Cyprus, CA, USA). A INAAT LOAD-based device have been designed for the detection of highly pathogenic avian influenza virus using a portable closed computer-controlled device [95]. A LOAD system has been developed for the detection of Rift Valley fever virus and yellow fever virus consisting of a low-cost, centrifugal microfluidic cartridge coupled with a lightweight and portable processing device [66,96] More recently, Hin et al. developed the FeverDisc, which is a fully integrated LOAD using LAMP technology which can detect the presence of *Plasmodium falciparum*, *P*. *vivax*, *P*. *ovale*, *P*. *malariae*; *Salmonella enterica* Typhi, *S*. *enterica* Paratyphi A, *Streptococcus pneumoniae*, chikungunya virus, dengue 1–4 and Zika virus. Hands-on time is around 5 min to add the sample into the cartridge with results generated in around 2 h using lyophilized amplification reagents, removing the need for cold chain logistics. The cost of the system has not been disclosed but is expected to be cheaper than current approaches [97].

### 6.3. Microfluidic Paper-Based Analytical Devices (µPADs)

The original method described in 2007 is a simple and inexpensive method that involves patterning paper with millimeter-sized channels [98]. Typical µPADs are composed of an arrangement of hydrophilic/hydrophobic microstructures deposited on paper that allow for the storage of reagents, sample movement through the µPAD, sorting, mixing and detection, all of which make µPADs particularly useful for field testing and other POC applications [90].

µPADs require no power source, are inexpensive to produce and are easy to transport; thus, they an ideal alternative to traditional techniques. µPADs have been adapted for the detection of Zika using NS1 protein and chikungunya viruses using a laser cut glass fiber µPAD that was capable of the detection of chikungunya-specific IgM in less than 10 min [99,100]. A µPAD device coupled to LAMP amplification has been hypothesized for the detection of SARS-CoV-2 using wax barriers to separate the individual chambers of the µPAD device. These chambers consist of a sample zone, buffer zone, LAMP master mix zone, mixing zone and finally a sensor zone [101]. Similar cost-effective and simple approaches could be applied to the detection of arboviral infection in POC applications.

### 6.4. Lateral Flow Devices

Lateral flow devices (LFDs) have been used in POC applications for many years and are probably the most common platforms used for the development of such assays. Their extensive adoption in POC development is a direct result of their low cost, ease of use, quicker results and simple interpretation. Applications of LFDs include heart disease [102], monitoring food toxins [103], food poisoning [104,105], bacterial infection [106], viral infection [107,108] and many more.

Although LFDs have been extensively used for many years, they generally lack the sensitivity and specificity of molecular methods, such as RT-PCR and INAAT. By incorporation of streptavidin and fluorescein labeled primers into the amplification reaction, lateral flow detection coupled with isothermal amplification has been applied to the HDA reaction for the detection of *T. vaginalis* [65] and biothreat agents [61]. LFDs have been combined with LAMP and RPA amplification for the detection of *Mycobacterium tuberculosis* [109], Influenza [110], Japanese encephalitis virus [111], Zika [112], SARS-CoV-2 [113], Monkeypox [114], African swine fever virus [115] and Heartland virus [116].

### 6.5. CRISPR-CAS12/13

CRISPR-Cas technology for pathogen detection relies on the inherent properties of CRISPR proteins. Clustered regularly interspaced short palindromic repeats (CRISPRs) are a family of proteins that provide an immune response in bacteria, which degrades foreign DNA guided by sequence specific RNA (crRNA) molecules [117]. The CRISPR-Cas9 system has been used extensively in biology for genome editing and the detection of DNA and RNA molecules due to the inherent specificity of the system. The method relies on two catalytic domains, RuvC and HNH, which induce site-specific cleavage at sequences complementary to the guide RNA [118]. The CRISPR-Cas12 system is distinct from the CRISPR-Cas9 family of proteins and uses a single catalytic domain, RuvC, to generate double-stranded DNA breaks at distinct sequences under the guidance of the crRNA. Cas12a enzymes recognize a T-rich protospacer adjacent motif (PAM) to generate double-stranded DNA breakage [119]. To produce a readable signal, ssDNA or RNA reporter molecules are added to the reaction, which can either be labeled with a biotin or fluorescent groups [117].

These assays have been paired with INAAT, such as LAMP, RPA, HDA and SDA, to produce assays that are cheap, deployable in the field and suitable for POC applications. To date, the CRISPR-Cas-12/13 systems have been used for the detection of several important arboviruses, including dengue, West Nile virus and Zika [86], Japanese encephalitis virus [120], Crimean-Congo hemorrhagic fever virus [121] and severe fever with thrombocytopenia symptoms virus [122].

Table 4 summarizes the main features of technologies used in arboviral diagnostics. Serological methods have been used for the diagnosis of arboviral disease for many years and have a proven track record, are generally easy to perform and are cost effective. However, these methods in general lack the sensitivity and specificity of molecular methods and can produce both false positive and false negative results.

NGS methods have the advantage that they can be used in viral surveillance programs to provide an unbiased screenshot, detecting any virus that is present in the clinical or environmental sample, which is very difficult to achieve using conventional molecular diagnostic techniques. NGS methods have a high level of sensitivity, which is again useful in viral surveillance programs to identify emerging pathogens in clinical samples. The major drawbacks of NGS are the high cost, dedicated infrastructure requirements and the reliance on highly trained technicians. Unlike serology, NGS technology is not applicable to POC and use in resource-limited settings.

RT-PCR is also a well-established technology that is highly sensitive and specific for the detection of specific pathogens in clinical and environmental samples. The technique does not have the high costs associated with NGS and has been adopted worldwide in hospital and pathology laboratories as a rapid method for pathogen detection. The drawbacks of RT-PCR include reliance on thermal cycling equipment and the requirement of trained staff to perform the assays. In general, RT-PCR is not compatible with POC, field deployable applications and resource-limited settings due to the expense of reagents and dependence on thermal cycling equipment.

INAAT is a newer technology that until recently has not achieved the widespread adoption of methods such as serology and RT-PCR. Like RT-PCR, INAAT has been shown to have a high level of sensitivity and specificity for the detection of target sequences. The method is rapid and can deliver results in less than 10 min in some cases. Unlike RT-PCR, the technology does not rely on a thermal cycler to achieve amplification; thus, it is more suited to POC and field deployable applications. INAAT has been combined with simple read-outs, such as naked eye visualization, LFD and other POC devices, resulting in assays that can be used in resource-limited settings.

## 7. Discussion

The vast majority of arboviruses are endemic in resource-limited countries and the burden of disease in those regions contributes to high levels of morbidity and mortality. Dengue is ubiquitous throughout the tropics, and it has been estimated that some 1–2 billion people are at risk of disease [123]. Over the past several decades, unprecedented geographical spread of dengue due to increased population growth, uncontrolled and unregulated urbanization, seasonal variations fueled by climatic change throughout equatorial tropical and subtropical regions has occurred. This has resulted in dengue becoming a major burden on resource-limited economies and their healthcare systems [124]. Vaccines against commonly found arboviruses have achieved some success in controlling diseases caused by some flavivirus family members. Yellow fever (YFV) and Japanese encephalitis (JEV) have largely become preventable diseases due to these efforts; however, effective vaccines against dengue [125] and Zika [126] are goals that are yet to be met.

As a result of the lack of specific vaccines and anti-viral drugs for many arboviral infections, early detection is critical to enable preventative measures during outbreaks of disease, such as vector control strategies, and to guide patient treatment regimens [123]. In addition, it has become pivotal for all economically underprivileged nations in tropical and sub-tropical regions to have effective surveillance mechanism combined with efficient preventative measures that are supported by strong epidemiological data to reduce the burden of disease [124]. This would help prevent epidemic transmission and the co-circulation of multiple serotypes of arboviruses such as dengue.

Under the ASSURED guidelines for resource-limited and POC settings, an assay should be Affordable, Sensitive, Specific, User friendly, Rapid and robust, Equipment free or simple and Deliverable. Although RDTs are simple to perform and relatively inexpensive, they lack specificity and sensitivity when compared to molecular diagnostic approaches. INAAT technologies conform to the ASSURED caveats; thus, they are ideal for use in the diagnosis of arboviral disease. The main advantages of these techniques over the more established RT-PCR methods are that they do not require expensive thermal cycling equipment, can be coupled to inexpensive read-outs such as lateral flow or naked eye detection, are more resistant to inhibitors that can be present in the primary sample and are generally faster than RT-PCR with time to results as little as 5 min.

Over the past 30 years, many INAAT technologies have been described, including NASBA, SDA, HDA and whole-genome amplification. Perhaps the two most widely adopted techniques are the LAMP and RPA methods which have been applied to a wide range of arboviral diseases. Both can generate results in less than 15 min and have been coupled with a range of simple detection methods, including LFDs, colorimetric, CRISPR-Cas12 and fluorescent-based technologies. Research is ongoing to produce enzymes that are capable of ambient temperature amplification, removing the requirement of a heat source and the lyophilization of reagents, eliminating the necessity of cold chain logistics and prolonging the viability of test kits. These improvements will greatly reduce costs involved and the ease with which they can be delivered and used in resource-limited settings and in the field. Lab on a chip and lab on a disc devices continue to be developed for POC and field deployable applications, thus minimizing the requirement of dedicated facilities and associated expenses to run them. With rapid developments in nanotechnology, microfluidics, miniaturization, 3D printing, prefabrication of components and the reduction in costs associated with these technologies, several devices have been constructed that only require the end user to add the primary sample. The simplicity of such devices mean that users will require limited formal training to perform assays removing the need for highly trained technicians and increasing the possibility of more convenient POC approaches.

During the SARS-CoV-2 pandemic, a plethora of research was carried out using the INAAT method for disease diagnosis. This greatly accelerated the acceptance of INAAT methods for use in routine molecular diagnostics. Many clinical trials were conducted comparing the sensitivity and specificity of INAAT to RT-PCR showing that these methods could be, in some cases, just as sensitive as more established techniques. Over the coming years, it is expected that new sample-to-answer INAAT-based devices will be produced that are cheaper to manufacture, easier to use, applicable as POC and deployable for the diagnosis of arboviral disease.

## Figures and Tables

**Table 1 microorganisms-11-01159-t001:** The table shows a number of important human arboviruses.

Virus	Family/Order	Vector	Symptoms	Reference
Zika virus (ZIKV)	*Flaviviridae*	* Aedes * mosquitoes	Fever, conjunctivitis, joint pain, headache, maculopapular rash, microcephaly, Guillain–Barré syndrome.	[10]
Yellow fever virus (YFV)	*Flaviviridae*	* Aedes * mosquitoes	Jaundice, liver damage, gastrointestinal bleeding, recurring fever.	[11]
Dengue Virus (DENV)	*Flaviviridae*	* Aedes * mosquitoes	Fever, headache, nausea, muscle and joint pain, skin rash, hypovolemic shock, hemorrhage.	[12]
West Nile virus (WNV)	*Flaviviridae*	*Culex* mosquitoes	Fever, headache, nausea, vomiting, swollen lymph nodes, meningitis, encephalitis, acute flaccid paralysis.	[13]
Japanese encephalitis virus (JEV)	*Flaviviridae*	*Culex* mosquitoes	Mild flu-like symptoms, encephalitis, seizures, paralysis, coma and long-term brain damage.	[14]
Tick-borne encephalitis virus (TBEV)	*Flaviviridae*	*Ixodes* ticks, *Dermacentor* and *Haemaphysalis*	Mild meningitis to severe meningoencephalitis with or without paralysis and long-term brain damage damage.	[15]
Omsk hemorrhagic fever virus (OHFV)	*Flaviviridae*	*Dermacentor* ticks	Fever, headache, nausea, muscle pain, cough and hemorrhages.	[16]
Saint Louis encephalitis virus (SLEV)	*Flaviviridae*	*Culex* mosquitoes	Headache, sensory depression, temporal–spatial disorientation, tremors and changes in consciousness.	[17]
Kyasanur Forest disease virus (KFDV)	*Flaviviridae*	*Haemaphysalis* *spinigera*	Fever with hemorrhagic and/or neurological features in 20% of patients.	[18]
Chikungunya virus (CHIKV)	*Togaviridae*	* Aedes * mosquitoes	Fever frequently associated with joint pain, polyarthralgia and arthritis, rash, myalgia and headache.	[19]
O’nyong nyong virus (ONNV)	*Togaviridae*	* Anophles * mosquitoes	Low-grade fever, symmetrical polyarthralgia, lymphadenopathy, generalized papular or maculopapular exanthema and joint pain.	[20]
Ross river virus (RRV)	*Togaviridae*	*Culex * and *Aedes* mosquitoes	Arthritis, rash, fever, fatigue and myalgia.	[21]
Eastern equine encephalitis virus (EEEV)	*Togaviridae*	* Culiseta * mosquitoes	Fever, chills, vomiting, myalgia, arthralgia, malaise and encephalitis.	[22]
Western equine encephalitis virus (WEEV)	*Togaviridae*	*Aedes, Culex* and *Culiseta* mosquitoes	Fever, chills, headache, aseptic meningitis and encephalitis.	[23]
Venezuelan equine encephalitis virus (VEEV)	*Togaviridae*	*Culex* mosquitoes	Fever, chills, malaise, severe headache, myalgia, seizures, drowsiness, confusion and photophobia.	[24]
Barmah Forest virus (BFV)	*Togaviridae*	*Culex* and *Aedes* mosquitoes	Asymptomatic to relatively mild symptomatic presentations, such as fever and rash; in more severe diseases, polyarthralgia or arthritis.	[25]
Thogoto virus (THOV)	*Orthomyxoviridae*	*Haemaphysalis* and *Amblyomma* ticks	Benign febrile symptoms to meningoencephalitis.	[26]
Rift Valley fever virus (RVFV)	*Bunyvirales*	*Culex* and *Aedes* mosquitoes	Fever, headache, backache, vertigo, anorexia, photophobia, hepatitis, jaundice, hemorrhagic disease and ocular complications	[27,28]
Ngari virus (NRIV)	*Bunyvirales*	*Aedes, Culex* and *Anopheles* mosquitoes	Fever, joint pain, rash, can induce severe and fatal hemorrhagic fever	[28]
Severe fever with thrombocytopenia syndrome virus (SFTSV)	*Bunyvirales*	*Haemophysalis* *Amblyomma*, *Ixodes* and *Rhipicephalus* ticks	High fever, gastrointestinal symptoms, thrombocytopenia, leukopenia and multiple organ failure	[29]
Crimean–Congo hemorrhagic fever virus (CCHFV)	*Bunyvirales*	*Hyalomma*, *Rhipicephalus* and *Dermacentor* ticks	Non-specific febrile illness, sudden onset of fever, myalgia, diarrhea, nausea and vomiting, hemorrhages at various sites around the body.	[30]
Jamestown Canyon virus (JCV)	*Bunyvirales*	*Aedes*, *Coquillettidia*, *Culex* mosquitoes	Non-specific febrile illness, meningitis or meningoencephalitis	[31]
La Crosse encephalitis virus (LACV)	*Bunyvirales*	* Aedes * mosquitoes	Fever, headache, myalgia, malaise and occasional prostration, encephalitis and lifelong sequelae.	[32]
Oropouche Virus (OROV)	*Bunyvirales*	*Culicoides* and *Culex* mosquitoes	Acute febrile illness, myalgia, arthralgia, dizziness, photophobia, rash, nausea, vomiting, diarrhea, conjunctive congestion and meningitis.	[33]

**Table 2 microorganisms-11-01159-t002:** The sequences of a selected region of a number of alphaviruses are shown before and after the 3base™ conversion process, which reduces the number of sequence variants from 575 before to only 27 after the conversion process.

	Sequence	
Alphavirus Species	Before Conversion	After Conversion
Barmah Forest virus	CCUUACUUCUGUGGAGGAUUU	TTTTATTTTTGTGGAGGATTT
Ndumu virus	CCGUAUUUCUGCGGCGGGUUC	TTGTATTTTTGTGGTGGGTTT
Chikungunya virus	CCUUACUUUUGUGGAGGGUUU	TTTTATTTTTGTGGAGGGTTT
O’nyong-nyong virus	CCAUACUUCUGUGGGGGAUUU	TTATATTTTTGTGGGGGATTT
Middelburg virus	CCCUACUUCUGCGGAGGGUUU	TTTTATTTTTGTGGAGGGTTT
Mayaro virus	CCCUACUUUUGUGGAGGUUUC	TTTTATTTTTGTGGAGGTTTT
Ross River virus	CCAUACUUCUGCGGCGGGUUU	TTATATTTTTGTGGTGGGTTT
Semliki Forest virus	CCAUAUUUUUGUGGGGGAUUC	TTATATTTTTGTGGGGGATTT
Una virus	CCUUACUUCUGCGGAGGAUUC	TTTTATTTTTGTGGAGGATTT
Aura virus	CCUUACUUUUGCGGCGGAUUU	TTTTATTTTTGTGGTGGATTT
Rio Negro virus	CCAUACUUUUGUGGAGGGUUU	TTATATTTTTGTGGAGGGTTT
Mucambo virus	CCGUACUUUUGCGGCGGGUUU	TTGTATTTTTGTGGTGGGTTT
Everglages virus	CCCUAUUUUUGUGGAGGGUUU	TTTTATTTTTGTGGAGGGTTT
Venezuelan equine encephalitis virus	CCCUAUUUUUGUGGAGGGUUU	TTTTATTTTTGTGGAGGGTTT
Eastern equine encephalitis virus	CCGUACUUUUGCGGAGGGUUC	TTGTATTTTTGTGGAGGGTTT
Western equine encephalitis virus	CCCUACUUCUGUGGGGGAUUU	TTTTATTTTTGTGGGGGATTT
Consensus sequence	CCNUAYUUYUGYGGDGGDUUY	TTDTATTTTTGTGGDGGDTTT
Number of variants	576	27

**Table 3 microorganisms-11-01159-t003:** Summary table comparing the main isothermal amplification techniques to RT-PCR.

Amplification Method	Template	Temperature	Primers	Time to Result	Enzymes (RNA)	Advantages	Disadvantages
Real-time PCR	DNA/RNA	Thermal cycling	2	15–60 min	2	Established method, high sensitivity and specificity; multiplexing.	Thermal cycler required and highly trained staff.
Nucleic-Acid-Sequence-Based Amplification	RNA	41 °C	2	90–120 min	3	Isothermal, rapid.	RNA-based amplification only.
Strand Displacement Amplification	DNA/RNA	37–65 °C	4	10–60 min	3	Isothermal, rapid results NEAR method <10 min. POC compatible	Sensitivity can be lower than other methods. Thermal denaturation required for DNA.
Helicase Dependant Amplification	DNA/RNA	37–65 °C	2	30–60 min	3	Isothermal, rapid. POC compatible.	Not as sensitive as other techniques
Loop mediated AMPlification	DNA/RNA	65 °C	4–6	5–30 min	2	Isothermal, rapid, POC adaptable. Multiplex capability.	Complex primer design.
Recombinase Polymerase Amplification	DNA/RNA	37–42 °C	2	5–30 min	3	Isothermal, rapid, POC compatible.	RUO only reagents available.

**Table 4 microorganisms-11-01159-t004:** Summary table comparing the four main technologies used in arboviral diagnosis.

Method	Cost	Ease of Use	Sensitivity	Specificity	POC Applicable	Advantages	Disadvantages
Serology	Low	Simple	Medium	Medium	Yes	Proven technology, cheap, easy to use, requires minimal infrastructure.	Sensitivity and specificity lower than nucleic acid detection technologies.
NGS	High	Highly complex	High	N/A, unless using targeted enrichment	No	High sensitivity provides unbiased results, ideal for surveillance approaches.	High cost, dedicated infrastructure. High level of technical skill required.
RT-PCR	Medium	Complex	High	High	No	Proven technology, high level of sensitivity and specificity.	Thermal cycler required. Not readily adaptable to POC
INAAT	Low-medium	Medium complexity	High	High	Yes	POC adaptable, no thermal cycling equipment required.	Can be complex to design, newer technology.

## Data Availability

The data presented in this study are available on request from the corresponding author.

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
