# Peer review of "Latest Advances in Arbovirus Diagnostics"

_microorganisms, 2023, doi:10.3390/microorganisms11051159_

Round 1

Reviewer 1 Report

Thank you for this review work which is necessary from time to time to clarify and take stock of these situations. My comments will be mainly on the form of your article because I think it can be improved and be more user friendly. 

Table 1: add the official abbreviation of the viruses. 

Figure 1: the map is difficult to read, maybe you could improve the aesthetics and put a logic in the colors (close colors when the same viruses co-circulate). Please add the meaning of the abbreviations in the legend. Are you sure that a representation by country is really relevant? In the Mediterranean area for example there are arboviroses circulating like Toscana and West Nile (at least) but on your map there doesn't seem to be any. In short, if you want to present a complete map of the world distribution of several arboviruses, it is better to take the time to do it correctly and seriously. Otherwise, you might as well not put a map at all rather than something that doesn't seem to be up to date anymore... Because as you rightly said, arboviruses evolve quickly and an obsolete map is useless. 

Please specify your references for arthropods belonging to Orthomyxoviridae and Poxviridae or give recent references. Otherwise, for my personal culture, you can tell me which viruses belonging to these families have been found to be transmitted by arthropods.

Concerning the paragraph on NGS: please elaborate a little bit on the sensitivity of the techniques.

Concerning the paragraph on RT-PCR, please elaborate a little bit more on the advantages and disadvantages of the new technique you present on bisultfite conversion. 

Suggestion: make a summary table for the different isothermal amplification methods.

Be careful with the numbering of the figures in the text. Figure 2 is difficult to read.

Table 2: adjust the size of the columns to the text. These data are interesting, but maybe you can also give us the number of different PCR or isothermal systems published for each virus.

Expand a bit more on your last paragraph about the use of CRISPR-cas-12/13? What is the difference with CRISPR-cas-9? Maybe add a diagram?

I find that your article lacks a visual summary, perhaps in the form of a table of the different techniques? I think it is important that people doing diagnosis can have a concentrate of information easily accessible.

Author Response

Thank you for your review of the manuscript which provided very useful points on how to improve the paper. I will try to answer all the comments you raised in the review.

Table 1: add the official abbreviation of the viruses.

I have added all official abbreviations to the Table 1. In addition, in response to Reviewer 2 who recommended that a number of columns should be removed to make the table easier to read and reduce the hyphenation I removed the distribution information which is now in figure 1 and case fatality rate.

Figure 1: the map is difficult to read, maybe you could improve the aesthetics and put a logic in the colors (close colors when the same viruses co-circulate). Please add the meaning of the abbreviations in the legend. Are you sure that a representation by country is really relevant? In the Mediterranean area for example there are arboviroses circulating like Toscana and West Nile (at least) but on your map there doesn't seem to be any. In short, if you want to present a complete map of the world distribution of several arboviruses, it is better to take the time to do it correctly and seriously. Otherwise, you might as well not put a map at all rather than something that doesn't seem to be up to date anymore... Because as you rightly said, arboviruses evolve quickly and an obsolete map is useless. 

I agree that figure 1 was rather difficult to read and not useful if it was not up to date so have used your suggestion to use close colours where a number of viruses circulate to make the map easier to follow. The meaning of the abbreviations have been added in the figure legend. I have also added information on West Nile virus and chikungunya circulation in the mediterranean region so that the map is more relevant and the distribution of the specified viruses up to date.

Please specify your references for arthropods belonging to Orthomyxoviridae and Poxviridae or give recent references. Otherwise, for my personal culture, you can tell me which viruses belonging to these families have been found to be transmitted by arthropods.

In the text I have now added information regarding Thogoto virus and Dhori viruses which are arboviruses belonging to the Orthomyxoviridae family and referenced the Gammaentomopoxvirus family containing arboviruses.

Concerning the paragraph on NGS: please elaborate a little bit on the sensitivity of the techniques.

In the text I have mentioned that NGS can in some cases have a similar sensitivity to RT-PCR and also added information on specific viral sequence enrichment strategies that can improve the sensitivity of NGS further.

Concerning the paragraph on RT-PCR, please elaborate a little bit more on the advantages and disadvantages of the new technique you present on bisultfite conversion. 

I have further elaborated on the bisulphite conversion method and added a table that makes the process easier to follow. I have also added that using this method it is possible to produce pan-family RT-PCR assays that can assist in arbovirus detection and surveillance.

Suggestion: make a summary table for the different isothermal amplification methods.

Your suggestion is a good one so I have added a new table that compares the different isothermal amplification methods and gives advantages and disadvantages of each technology compared to conventional RT-PCR.

Be careful with the numbering of the figures in the text. Figure 2 is difficult to read.

I have corrected the numbering issues in the text. Figure 2 has been reduced to include only SDA, HDA, LAMP and RPA which makes the figure easier to read and with NASBA I have added a full explanation of the method in the relevant section. Also the figure has now been fully referenced.

Table 2: adjust the size of the columns to the text. These data are interesting, but maybe you can also give us the number of different PCR or isothermal systems published for each virus.

Table 2 has been removed as reviewer 2 thought that it added nothing to the manuscript. I am happy to revisit this if you wish.

Expand a bit more on your last paragraph about the use of CRISPR-cas-12/13? What is the difference with CRISPR-cas-9? Maybe add a diagram?

I have expanded on the paragraph regarding the CRISPR technology and explained the differences between the CRISPR-cas9 and CRISPR-cas12/13 technologies which I hope will suffice without a diagram. Reference 119 does provide excellent reading on the technology.

I find that your article lacks a visual summary, perhaps in the form of a table of the different techniques? I think it is important that people doing diagnosis can have a concentrate of information easily accessible.

I have added a new table which gives a summary of the different methods used in arbovirus diagnostics including serology, NGS, RT-PCR and INAAT and lists the advantages and disadvantages of each.

Reviewer 2 Report

The review of the “Latest Advances in Arbovirus Diagnostics” authored by Millar et al.  covered current testing technology in the expanding filed of Arbovirology.  The review strived to include old proven techniques along with newer and still unproven assays in the clinical setting.  Weighing in on the advantages and disadvantages in regard to the ability to implement each assay in a point of care setting as well as an impoverished nation.  The authors concentrated on the cost and need for modern technology used in the current accepted clinical assays while trying to introduce those technologies that could possibly overcome those hurdles.

Writing a comprehensive review of such a broad set of data is a complicated process to get balanced between what the author is trying to introduce as new versus what is the established in the field.  Throughout the paper there were instances of sentences that never quite conveyed their intended point. Line 32-33.  And repetition of phrases such as “may be” between lines 71 and 78 that caused me as a reader to start questioning the content of the paper.  The idea of the paper was a good one but one that was lost in the writing style and figures.

The first item that really stood out was table 1.  The narrow columns that forced so much of the table to be hyphenated made what was being shown very hard to understand.  A reworking of this table in a manner that eliminates at least 1 or 2 columns is needed.  Perhaps grouping the Flavi viruses together and then followed by Toga and so on would help.  And the information under Case fatality rate is mostly presented in the expected percentages but the very first entry is not a fatality rate at all.  The distribution can be more general, which might prevent one entry from blending into the next.

In line 101 the authors state that a “small number of arboviruses cause human disease” followed by a number of sentences directly countering this statement.   This greatly weakens the paper’s message.  Lines 110-113 repeat the statements made in lines 64-66.  Line 118-120 list a number of assays but in a manner that makes it difficult to follow which assay follows the previous and should be rewritten.  Line 135 needs a comma after “method”. 

Lines 141-152 present a number of instances that NGS was used to identify viruses but simply lists them sentence after sentence without a real conclusion or purpose, this can be fixed.  In line 166 the author introduces a “novel technology” but does not provide enough information to understand it and it is from the authors company so it simply looks like a random plug for their assay. 

Line 174 there is no such thing as “Miniaturized PCR”  what they describe is cassette based testing using RT-PCR technology.  And Veredus assay they describe is for research use only which may or may not fit in with the intent of the review.

Section 4.2 on line 207 is lacking in clarity of how NASBA really works, the information provided does not include that it is predominantly an RNA based system.  The next section for HDA provides some information that the helicase unwinds the DNA but never makes the direct statement that it is this process that eliminates the need for the thermocycling used in traditional PCR. 

The paper repeatedly refers to Figure 1 but should refer to Figure 2.  It was at this point that the conclusion was made that this paper was submitted without adequate review by the authors.  Figure 2 is too small to be of use in a printed article and the title is in an incorrect font or alignment as well.  The figures need to be referenced or it needs to be made clear what was used to generate them for this review.  Table 2 provides no additional information to the review and should be eliminated.  And Figure 3 which is a table in line of the text and not labelled as a table should not be centered and probably could be eliminated.  The ASSURED and REASSURED programs can be presented in another manner. 

It is my opinion that everyone knows what a Lateral flow device is in the scientific community and the reference to a pregnancy test with a weblink is entirely not needed in any paper or review. 

Author Response

Thank you for reviewing the paper. Your criticisms were most useful and I agree that the paper needed further review and correction to a number of sentences as they did not easily convey the intended message. I have attempted to address all your points and correct each in the manuscript.

Throughout the paper there were instances of sentences that never quite conveyed their intended point. Line 32-33.  And repetition of phrases such as “may be” between lines 71 and 78 that caused me as a reader to start questioning the content of the paper.  The idea of the paper was a good one but one that was lost in the writing style and figures.

I have corrected these sentences and a number of others in the manuscript that contained repetition and did not easily convey the intended message.

The first item that really stood out was table 1.  The narrow columns that forced so much of the table to be hyphenated made what was being shown very hard to understand.  A reworking of this table in a manner that eliminates at least 1 or 2 columns is needed.  Perhaps grouping the Flavi viruses together and then followed by Toga and so on would help.  And the information under Case fatality rate is mostly presented in the expected percentages but the very first entry is not a fatality rate at all.  The distribution can be more general, which might prevent one entry from blending into the next.

I agree that table 1 was difficult to read due to the repeated hyphenation of words resulting in one entry blending into the next. To address this the distribution and case fatality columns have been removed in order to produce a more easily readable table. The distribution of selected viruses is now found only in figure 1 which has also been revised. 

In line 101 the authors state that a “small number of arboviruses cause human disease” followed by a number of sentences directly countering this statement.   This greatly weakens the paper’s message.  Lines 110-113 repeat the statements made in lines 64-66.  Line 118-120 list a number of assays but in a manner that makes it difficult to follow which assay follows the previous and should be rewritten.  Line 135 needs a comma after “method”. 

These issues have all been corrected with line 101 now starting with "A significant number of arboviruses cause human disease" and the repeat statements have been removed. Lines 118-120 have been modified and rewritten and the comma added after method.

Lines 141-152 present a number of instances that NGS was used to identify viruses but simply lists them sentence after sentence without a real conclusion or purpose, this can be fixed.

This has been rewritten and the method explained more fully for the applications listed and a conclusion added.

In line 166 the author introduces a “novel technology” but does not provide enough information to understand it and it is from the authors company so it simply looks like a random plug for their assay. 

A new table and wording has been added to explain the technology further which was also requested by reviewer 1. The technology is of particular relevance to arbovirus diagnostics and surveillance (reference 54) thus was added as it is a new RT-PCR method which could have an impact. However, if it is not appropriate to add in the RT-PCR section it can be removed.

Line 174 there is no such thing as “Miniaturized PCR”  what they describe is cassette based testing using RT-PCR technology.  And Veredus assay they describe is for research use only which may or may not fit in with the intent of the review.

This has been changed to sample to answer RT-PCR systems and the RUO status of the Veredus system included.

Section 4.2 on line 207 is lacking in clarity of how NASBA really works, the information provided does not include that it is predominantly an RNA based system.  The next section for HDA provides some information that the helicase unwinds the DNA but never makes the direct statement that it is this process that eliminates the need for the thermocycling used in traditional PCR. 

The section on NASBA has been rewritten to provide a full explanation of the method including that it is a RNA based system. This information has also been included in a new table, suggested by reviewer 1, briefly summarising the isothermal methods and providing the advantages and disadvantages of each. The HDA section has also been modified as suggested.

The paper repeatedly refers to Figure 1 but should refer to Figure 2.  It was at this point that the conclusion was made that this paper was submitted without adequate review by the authors. 

This has been corrected.

Figure 2 is too small to be of use in a printed article and the title is in an incorrect font or alignment as well.  The figures need to be referenced or it needs to be made clear what was used to generate them for this review. 

I agree Figure 2 was too small and in order to improve this the descriptions of WGA and NASBA have been removed to concentrate on SDA, HDA, LAMP and RPA. The NASBA method is described in detail in the text. References and the method used to generate the figure have been included.

Table 2 provides no additional information to the review and should be eliminated.  And Figure 3 which is a table in line of the text and not labelled as a table should not be centered and probably could be eliminated.  The ASSURED and REASSURED programs can be presented in another manner. 

Table 2 has been removed as has figure 3 and the ASSURED/REASSURED programs discussed in the text.

It is my opinion that everyone knows what a Lateral flow device is in the scientific community and the reference to a pregnancy test with a weblink is entirely not needed in any paper or review. 

This unfortunate inclusion has been removed. A table summarising the advantages and disadvantages of different diagnostic techniques including serology, RT-PCR, NGS and INAAT has also been added at the request of reviewer 1.

Round 2

Reviewer 2 Report

After the edits were made the paper is OK to publish.